# Application of Novel AC–AC Matrix VFD for Power Factor Improvement in Conventional AC–DC–AC VFD-Loaded Power Distribution Lines

Gytis Petrauskas [1,*], Gytis Svinkunas [2], Audrius Jonaitis [2] and Andreas Giannakis [3]

1 Department of Automation, Faculty of Electrical and Electronics Engineering, Kaunas University of Technology, LT-51394 Kaunas, Lithuania
2 Department of Electric Power Systems, Faculty of Electrical and Electronics Engineering, Kaunas University of Technology, LT-51394 Kaunas, Lithuania; gytis.svinkunas@ktu.lt (G.S.); audrius.jonaitis@ktu.lt (A.J.)
3 Department of Electric Power Engineering, Norwegian University of Science and Technology, 7491 Trondheim, Norway; andreas.giannakis@ntnu.no
* Correspondence: gytis.petrauskas@ktu.lt; Tel.: +370-687-17526

**Abstract:** In this study, an innovative approach to matrix-converter-based AC—AC variable frequency drives (VFDs) is introduced. The possibility of using AC–AC matrix VFDs for reactive power compensation in conventional AC–DC–AC VFD-loaded power distribution lines is investigated. It is found that the interaction of a large number of conventional AC–DC–AC VFDs with a conventional capacitor-based local compensation device leads to overcompensation in 0.4 kV power distribution lines. This is due to the fact that the conventional compensation device is designed to compensate the lagging reactive power produced by inductive loads, such as AC motors. This highlights the demand for the compensation of leading reactive power that is not predicted by the designer. To solve this problem, the modification of a certain number of previously installed VFDs by replacing their conventional AC–DC–AC converters with AC–AC matrix converters is proposed. This can lead to improvements in the power factor in 0.4 kV power distribution lines. In this study, the range of reactive power produced by conventional AC–DC–AC VFDs was determined mathematically, by simulation, and experimentally. The range of reactive power produced by the novel AC–AC matrix VFD was also determined. On that basis, the number of VFDs to be modified is defined to keep the power factor close to unity.

**Keywords:** matrix converter; space vector modulation; power factor; variable frequency drive



## 1. Introduction

Power generation and transmission are complex processes that require the interconnected operation of the components of the power system. The reactive power is one of the main components in the system. Loads such as electric motors require reactive power for their operation. To improve the performance of the AC power systems, this reactive power has to be managed. This is known as reactive power compensation [1].

Typically, the current in old-fashioned power distribution lines (PDL) lags behind voltage, because of inductive loads such as motors. Local reactive power compensation devices are usually designed to compensate the lagging reactive power [2].

The evolution of variable frequency drive (VFD) technology has changed the type of reactive power in 0.4 kV PDL. VFDs are finding increasing applications in various industrial and infrastructure sectors. The global VFD market size is expected to be 31.3 billion USD by 2025, from an estimated 19.2 billion USD in 2020, rising at a market growth of 6% for the mentioned forecast period.

A typical induction motor connected directly to a power distribution line is a lagging reactive power source. When connecting this motor via an AC–DC–AC frequency converter,

a conventional VFD is formed. This VFD, unlike the induction motor, produces the leading reactive power. Data on the reactive power produced by VFDs obtained experimentally are given below. Reactive power compensators, traditionally designed to compensate the inductive (lagging) reactive power, are not capable of compensating the leading reactive power produced by the VFD. It carries financial losses. Companies that are unable to compensate the VFD-produced reactive power pay fines. Some studies have suggested solutions to this problem. The authors of [3] proposed to compensate the reactive power using reactive elements. The authors of [4] defined power quality coefficients that determine the distortion caused by variable-speed asynchronous electric drives. The authors of [5] suggested the use of dynamic reactive power compensation equipment.

Recently, VFDs based on a novel matrix converter (MC) are becoming more widespread in buildings and industry. With the development of semiconductor technologies, the popularity of MC is constantly increasing. In the AC–AC matrix VFD, indirect space vector modulation (ISVM) is the most common way to control converter switches. Using this control method, the MC is virtually divided into two parts. The first part is the current source rectifier (CSR). The second part is the voltage source inverter (VSI). This division makes it possible to control the current displacement angle independently of the voltage on the motor side. Such a VFD connected to the PDL can control the reactive power at the connection point. With regard to the method of control mentioned above, the MC-based induction motor VFD has the ability to exchange both leading and lagging reactive power with the connected power distribution line. This means that the AC–AC matrix VFD can be used for conventional AC–AC–AC VFD-produced reactive power compensation. It is, therefore, proposed to replace the electronic equipment of one of the conventional VFDs installed in a commercial or industrial building. Instead of three stages (diode supply-side rectifier, intermediate DC circuit, and inverter), nine bidirectional electronic switches are installed, leaving the VFD induction motor the same. In this way, a PDL power factor close to unity can be achieved.

Some authors of the reviewed publications suggested using ISVM-controlled MC for reactive power compensation. The authors of [6] suggested the use of a permanent magnet machine powered by a MC as the compensation device. The authors of [7] proposed to connect the AC–AC matrix VFD to the PDL, thus compensating for the reactive power produced by light-emitting diode light sources. The authors of [8] proposed using an MC in combination with a transformer for power factor control. The authors of [9,10] presented a modulation strategy allowing the expansion of the MC reactive power range. The authors of [11] suggested the use of AC–AC matrix VFDs for voltage sag impact reduction. The only study that considered the reactive power in the induction motor MC VFD was [12]. However, the authors of this publication only proposed a control strategy that provides current control and a unity power factor. They did not measure the ability to compensate for the reactive power produced by other sources, such as conventional AC–DC–AC VFDs. Therefore, a comprehensive study of MC-based induction motor VFDs is needed to determine the range of reactive power compensation they provide before they can be put into practical application. The number of conventional AC–DC–AC VFDs whose reactive power can be compensated for by a single AC–AC matrix VFD must be determined. Furthermore, the effect of motor load torque and rotor speed on the reactive power of conventional AC–DC–AC VFDs must be investigated.

## 2. Overcompensation Problem in Current Power Distribution Lines

Typically, the current in conventional PDLs lags behind voltage, because of inductive loads such as AC motors (Figure 1a).

However, currently, as shown in Figure 1b, more and more AC motors are powered by AC–DC–AC frequency converters. One component of the conventional VFD directly connected to the PDL is the diode rectifier. As shown in Figure 2, another large and sensitive element is the electrolytic capacitor. The value of this capacitor is usually designed as 85 µF/kW. The range of voltage variation in such a case is 5–10% of the nominal voltage.

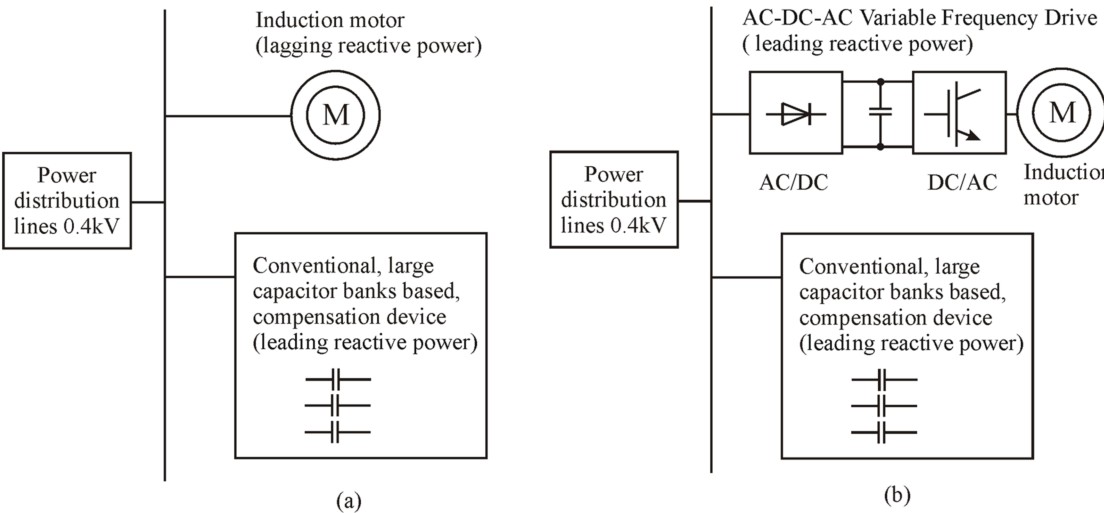

**Figure 1.** The loads and compensation devices of a 0.4 kV PDL: (**a**) conventional load equipment; (**b**) current load equipment (causes overcompensation).

### 2.1. Theoratical Analysis of the Reactive Power Generated by Diode Rectifier Supply-Side Variable Frequency Drive

The reactive power of such equipment directly depends on the voltage at the power distribution line connection point and the current of the diode rectifier (Figure 2).

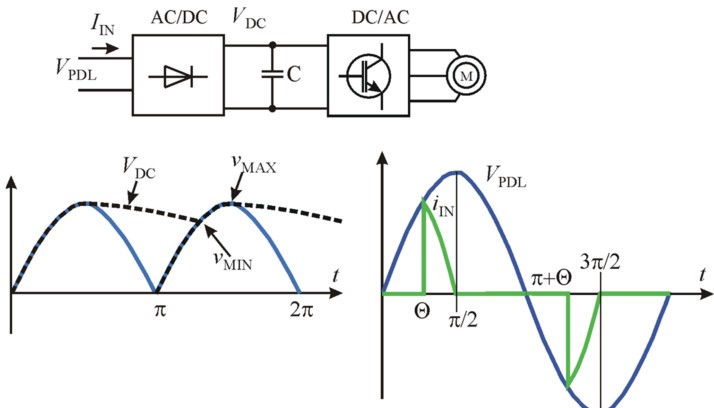

**Figure 2.** Time diagrams of the voltage and current of single-phase diode rectifier supply-side VFD.

The input current of the AC–DC device, due to the large number of harmonics, is quite complex and can be described by the following Equation (1) [7]:

$$I_{IN}(t) = I_0 + I_1 \sin(\omega t + \varphi_1) + I_2 \sin(\omega t + \varphi_2) \cdots \tag{1}$$

The consumption or generation of reactive power depends on $I_1 \sin(\omega t + \varphi_1)$. Because of this, it must be expressed in a Fourier series, and $\varphi_1$ must be determined as follows:

$$\tan \varphi_1 = \frac{I_1''}{I_1'} = \frac{(\pi - 2\theta - \sin 2\theta)}{(1 + \cos 2\theta)}. \tag{2}$$

In the case of $\theta_1 = 64°$, $\varphi_1 = 19.1°$ when $k_V = 10\%$, $\tan \varphi_1 = 0.34$. In the case of $\theta_2 = 72°$, $\varphi_2 = 16.1°$ when $k_V = 5\%$, $\tan \varphi_2 = 0.28$. These are the minimum and maximum limits of the current displacement angle. Since $\varphi_1 > 0$ and $\varphi_2 > 0$, it can be stated that the conventional AC–DC–AC VFD injects some amount of the leading reactive power into the PDL. It injects 0.28–0.34 kVar in the case of powering a 1 kW load.

For more powerful VFDs, a three-phase diode rectifier supply is used (Figure 3). An equation for the expression of the reactive power on the forward-conducting angle of the diodes can be derived for this rectifier.

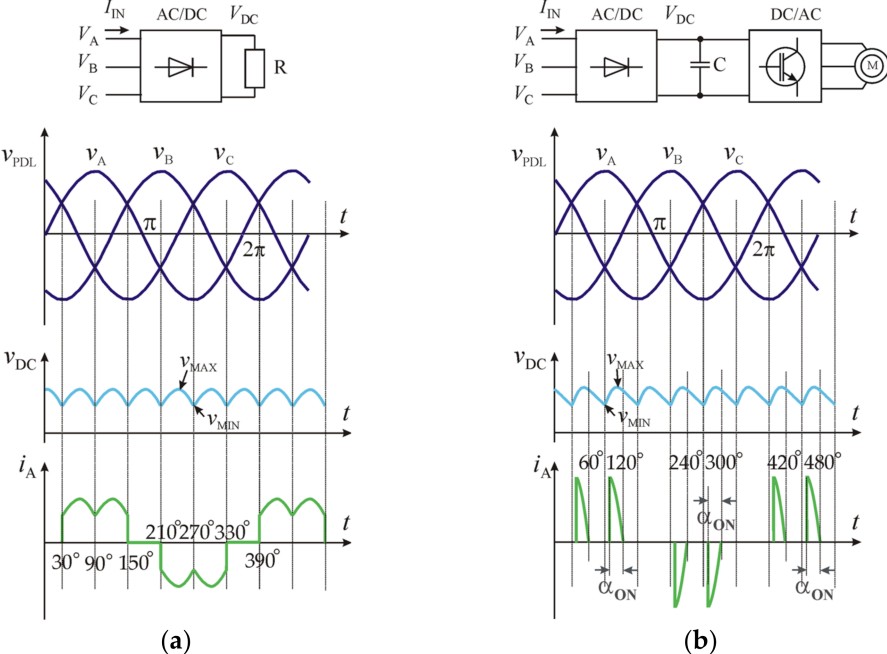

(a)  (b)

**Figure 3.** Time diagrams of the voltage and current of three-phase diode rectifier supply-side VFD: (**a**) pure resistor load; (**b**) conventional VFD.

The voltage pulsations and current of diode rectifier-based devices can be described by the following common equations:

$$V_A = V_m sin\omega t, \tag{3}$$

$$i_{IN} \approx C\frac{dV_A}{dt} = C\ V_m\ \omega\ cos\omega t. \tag{4}$$

$$\Delta V_{PULS} = V_{MAX} - V_{MIN}, \tag{5}$$

$$k_V = \frac{\Delta V_{PULS}}{V_{MAX}}. \tag{6}$$

According to the technical requirements, the voltage pulsation $k_V$ on the DC capacitor is in the range of 5–10%. According to this, angle $\alpha_{ON}$ determines the moment forward-conduction of the VFD's supply-side diode rectifier.

$$V_{DC} = V_A - V_B = V_m sin\omega t - V_m sin(\omega t - 120°) = 2cos\frac{\omega t+(\omega t-120°)}{2} \times sin\frac{120°}{2} = \\ 2cos(\omega t - 60°) \times \left(\frac{\sqrt{3}}{2}\right) = \sqrt{3}cos(\omega t - 60°) = \sqrt{3}sin(\omega t + 30°). \tag{7}$$

The maximum value is obtained when $\omega t = 60°$ or $120°$, where $sin\ (60° + 30° - \alpha_{ON}) = 0.9$–$0.95$. Therefore,

$$\alpha_{ON} = 26°, \ at\ k_V = 10\%;$$

$$\alpha_{ON} = 18°, \ at\ k_V = 5\%;$$

$$\alpha_{ON} = 13°, \ at\ k_V = 2,5\%.$$

The type of reactive power (leading or lagging) is determined by the component $i_1 = I_{1m}sin(\omega t + \varphi_1)$ in the Fourier series (Equation (1)).

$$I_1' = \frac{1}{\pi} \int_{60°-\alpha_{ON}}^{60°} cos(\omega t + 30°)sinxdt\omega t + \frac{1}{\pi} \int_{120°-\alpha_{ON}}^{120°} cos(\omega t - 30°)sinxdt\omega t + \frac{1}{\pi} \int_{240°-\alpha_{ON}}^{240°} cos(\omega t + 30°)sinxdt\omega t + \frac{1}{\pi} \int_{300°-\alpha_{ON}}^{300°} cos(\omega t - 30°)sinxdt\omega t. \tag{8}$$

$$I_1'' = \frac{1}{\pi} \int_{60°-\alpha_{ON}}^{60°} cos(\omega t + 30°)cosxdt\omega t + \frac{1}{\pi} \int_{120°-\alpha_{ON}}^{120°} cos(\omega t + 30°)cosxdt\omega t + \frac{1}{\pi} \int_{240°-\alpha_{ON}}^{240°} cos(\omega t + 30°)cosxdt\omega t + \frac{1}{\pi} \int_{300°-\alpha_{ON}}^{300°} cos(\omega t - 30°)cosxdt\omega t. \tag{9}$$

$$I_1' = \frac{1}{2\pi}\left[\sqrt{3} + cos(150° - 2\alpha_{ON}) - cos(30° - 2\alpha_{ON})\right]. \tag{10}$$

$$I_1'' = \frac{1}{2\pi}\left[\frac{4\sqrt{3}}{2} - sin(150° - 2\alpha_{ON}) - sin(210° - 2\alpha_{ON})\right] \tag{11}$$

$$tan\,\varphi_1 = \frac{\frac{1}{2\pi}\left(2\sqrt{3}\alpha_{ON} - sin(150° - 2\alpha_{ON}) + sin(30° - 2\alpha_{ON})\right)}{\frac{1}{2\pi}\left(\sqrt{3} - cos(150° - 2\alpha_{ON}) - cos(30° - 2\alpha_{ON})\right)}. \tag{12}$$

According to Equation (12),

$$\alpha_{ON} = 13°, \; k_V = 2,5\%, \; \varphi_1 = 8.86°, \; tan\,\varphi_1 = 0.152,$$
$$\alpha_{ON} = 18°, \; k_V = 5\%, \; \varphi_1 = 11.98°, \; tan\,\varphi_1 = 0.22,$$
$$\alpha_{ON} = 26°, \; k_V = 10\%, \; \varphi_1 = 17.27°, \; tan\,\varphi_1 = 0.311.$$

The results of the calculation show that, at the same pulsation level, the three-phase rectifier generates less reactive power than the single-phase rectifier.

*2.2. Experiemntal Analysis of the Reactive Power Generated by Diode Rectifier Supply-Side Ariable Frequency Drive*

The experimental analysis of the reactive power in the VFD loaded PDL was performed with a METREL MI 2892 power quality analyzer. That instrument records all measurement (Classic and Modern), regardless of selected method. This device complies with the measurement methods defined in the IEEE 1459 standard. The old measurement methods were applicable when currents and voltages were close to sinusoidal ones. However, they are not suitable for modern measurements when we have power electronics equipment such as VFD representing a nonlinear load for industrial and commercial PDL. The methodology used in this study measures the fundamental and nonfundamental components separately, as shown in Figure 4.

The experimentally obtained results of power analysis are presented in Table 1 (single-phase supply AC-DC-AC VFD) and in Table 2 (three-phase supply AC-DC-AC VFD). The following power distribution line voltage parameters were investigated:

- $P_C$—active power (combined) of all voltage and current harmonic;
- $N_C$—reactive power (combined) of all voltage and current harmonic
- $S_C$—apparent power (combined) of all harmonic
- $PF$—total effective power factor (combined) of all harmonics;
- $P_{Fund}$—active power of fundamental harmonics;
- $Q_{VFund}$—reactive power of fundamental harmonics;
- $S_{VFund}$—apparent power vector of fundamental harmonics;
- $PF_{VFund}$—displacement factor vector of fundamental harmonics.

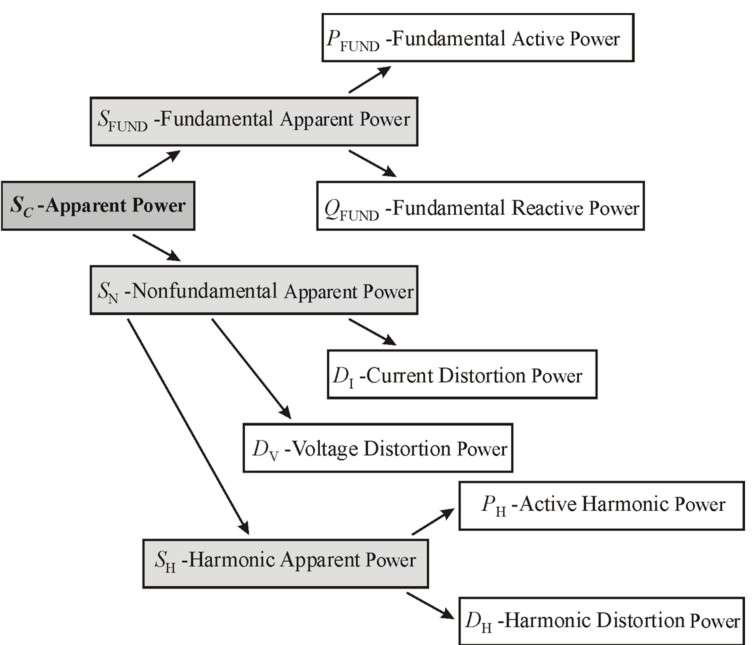

**Figure 4.** The power management organization according to the IEEE1459.

**Table 1.** Power components at the point of connection of Siemens Micromaster 420 single-phase diode rectifier supply-side VFD to the PDL, keeping motor load torque constant.

| $f_{OUT}$, Hz | Combined | | | | Fundamental | | | | |
|---|---|---|---|---|---|---|---|---|---|
| | $P_C$, W | $N_C$, var | $S_C$, VA | $PF_C$ | $P_{Fund}$, W | $Q_{VFund}$, var | $S_{VFund}$, VA | $PF_{VFund}$ | $tan(\varphi)$ |
| 10 | 32.8 | −58.01 | 58.1 | 0.565 | 32.6 | −16.5 | 36.9 | 0.894 | −0.5 |
| 20 | 40.9 | −71.4 | 71.7 | 0.571 | 40.7 | −16.7 | 44.5 | 0.927 | −0.405 |
| 30 | 50.1 | −82.5 | 84.1 | 0.596 | 49.9 | −16.6 | 53.3 | 0.950 | −0.328 |
| 40 | 60.4 | −95.5 | 98.4 | 0.614 | 60.2 | −17.0 | 63.6 | 0.963 | −0.278 |
| 50 | 67.6 | −106 | 109 | 0.618 | 67.4 | −17.6 | 70.8 | 0.968 | −0.256 |

**Table 2.** Power components at the point of connection of Rockwell Automation Power Flex 70 three-phase diode rectifier supply-side VFD to the PDL, keeping motor load torque constant.

| $f_{OUT}$, Hz | Combined | | | | Fundamental | | | | |
|---|---|---|---|---|---|---|---|---|---|
| | $P_C$, W | $N_C$, var | $S_C$, VA | $PF_C$ | $P_{Fund}$, W | $Q_{VFund}$, var | $S_{VFund}$, VA | $PF_{VFund}$ | $tan(\varphi)$ |
| 10 | 175 | −438 | 472 | 0.3705 | 171 | −102 | 200 | 0,858 | −0.596 |
| 20 | 257 | −585 | 640 | 0.4021 | 253 | −103 | 273 | 0.925 | −0.407 |
| 30 | 342 | −730 | 807 | 0.424 | 336 | −104 | 352 | 0.954 | −0.311 |
| 40 | 433 | −857 | 961 | 0.450 | 427 | −100 | 439 | 0.973 | −0.235 |
| 50 | 530 | −997 | 1131 | 0.469 | 522 | −108 | 534 | 0.978 | −0.208 |

The forward-conducting and reverse-blocking angles of the VFD's supply-side diode were determined from the experimental voltage and current time diagrams presented in Figure 5. These data are consistent with the theoretical assumptions (Figure 2) and with the data from the mathematical calculations performed according to Equation (2). According to the experimental data, the forward-conducting angle was 72.6°, leading to the calculation of $tan(\varphi) = 0.28$. A similar result can be obtained from the ratio of the reactive power of fundamental harmonics to the active power of fundamental harmonics measured experimentally using a power quality analyzer: $tan(\varphi) = Q_{VFund}/P_{Fund} = 17.6/67.4 = 0.26$.

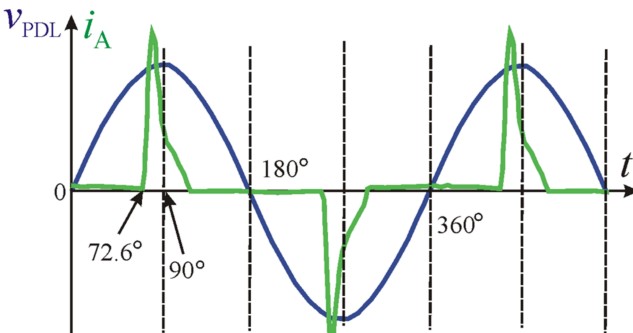

**Figure 5.** Current and voltage time diagrams at the point of connection of Siemens Micromaster 420 single-phase diode rectifier supply-side VFD to the PDL, with a motor supply voltage frequency of 50 Hz.

The forward-conducting and reverse-blocking angles of the VFD's supply-side diode were determined from the experimental voltage and current time diagrams presented in Figure 6. These data confirm the theoretical assumptions presented in Figure 3. These data were also compared with the data of the mathematical calculations performed according to Equation (12). A similar result can be obtained from the ratio of the reactive power of fundamental harmonics to the active power of fundamental harmonics measured experimentally using a power quality analyzer: $\tan(\varphi) = Q_{\text{VFund}}/P_{\text{Fund}} = 108/522 = 0.20$. Thus, the ratio is higher than the theoretical one.

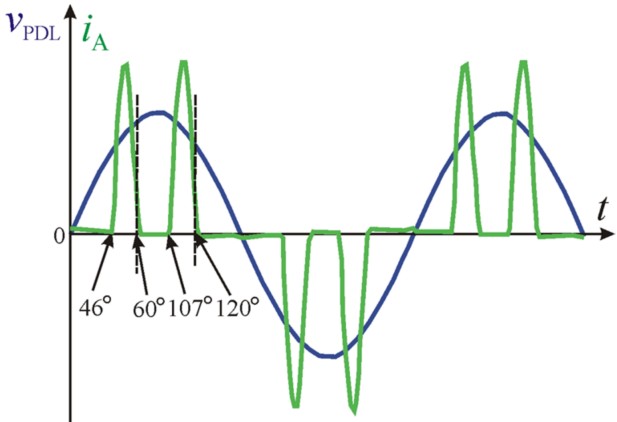

**Figure 6.** Current and voltage time diagrams at the point of connection of Rockwell Automation PowerFlex 70 three-phase diode rectifier supply-side VFD to the PDL, with a motor supply voltage frequency of 50 Hz.

The reactive power produced by AC–DC–AC VFDs cannot be compensated by using a conventional capacitor-based local compensation device. In this case, a universal compensation device is required, which has the ability to compensate for both types of reactive power: leading and lagging.

Inductive reactors designed to reduce harmonic distortion can perform the function of compensating the reactive power produced by VFDs. However, this method of compensation has significant drawbacks. The first disadvantage is the cost of compensation equipment due to the high cost of ferrous metals. Another drawback is the high energy loss. A third disadvantage is the high weight and dimensions of the compensation equipment. These drawbacks are evident in Table 3, which presents the data for mass-produced reactive power compensation reactors.

The use of an induction motor powered by an MC as the universal compensation device was researched. Depending on the structure of the converter and the specific controls, it can be said that AC–AC matrix VFDs may be suitable for both leading and lagging reactive power compensation.

**Table 3.** Inductive reactors of Schneider electric designed for 400 V, 50 Hz PDL.

| Relative Impedance, % | Capacitor Power, kVar | Inductance, mH | IMP, A | Reactor Power, kVar | Max Losses, kW | Power Losses in Percent of Reactor Power, % | Weight, kg | Reference Number |
|---|---|---|---|---|---|---|---|---|
| 4% | 6.5 | 11.439 | 10 | 1.07 | 0.1 | 9.3 | 10 | LVR14065A40T |
| | 12.5 | 6.489 | 20 | 2.44 | 0.15 | 6.14 | 15 | LVR14125A40T |
| | 25 | 3.195 | 40 | 4.80 | 0.2 | 4.16 | 22 | LVR14250A40T |
| | 50 | 1.598 | 80 | 9.58 | 0.4 | 4.17 | 33 | LVR14500A40T |
| | 100 | 0.799 | 160 | 19.09 | 0.6 | 3.14 | 55 | LVR14X00A40T |

## 3. Special Features of the AC–AC Matrix VFD

The researched compensation device consisted of an MC, whose input was connected to the PDL and powered the induction motor (Figure 7). The MC used the forced switching of nine bidirectional switches. These nine switches generated the supply voltage to the motor stator windings of the required frequency. Unlike conventional AC–DC–AC VFDs, the AC–AC matrix VFD contains no capacitors. These nine bidirectional switches created combinations of 27 active, passive, and zero vectors, as shown in Table 4.

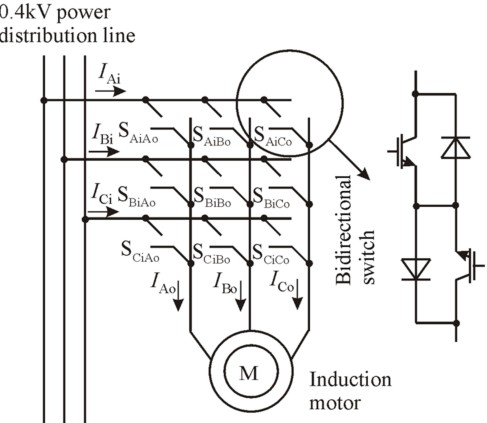

**Figure 7.** Functional diagram of MC-based induction motor VFD.

**Table 4.** Active combinations created by nine bidirectional switches.

| Vector. | Ao | Bo | Co | $S_{AiAo}$ | $S_{BiAo}$ | $S_{CiAo}$ | $S_{AiBo}$ | $S_{BiBo}$ | $S_{CiBo}$ | $S_{AiCo}$ | $S_{BiCo}$ | $S_{CiCo}$ |
|---|---|---|---|---|---|---|---|---|---|---|---|---|
| −3 | Ai | Ci | Ci | 1 | 0 | 0 | 0 | 1 | 0 | 0 | 0 | 1 |
| +2 | Bi | Ci | Ci | 1 | 0 | 0 | 0 | 0 | 1 | 0 | 0 | 1 |
| −1 | Bi | Ai | Ai | 0 | 1 | 0 | 1 | 0 | 0 | 1 | 0 | 0 |
| +3 | Ci | Ai | Ai | 0 | 0 | 1 | 1 | 0 | 0 | 1 | 0 | 0 |
| −2 | Ci | Bi | Bi | 0 | 0 | 1 | 0 | 1 | 0 | 0 | 1 | 0 |
| +1 | Ai | Bi | Bi | 1 | 0 | 0 | 0 | 1 | 0 | 0 | 1 | 0 |
| −6 | Ci | Ai | Ci | 0 | 0 | 1 | 1 | 0 | 0 | 0 | 0 | 1 |
| +5 | Ci | Bi | Ci | 0 | 0 | 1 | 0 | 1 | 0 | 0 | 0 | 1 |
| −4 | Ai | Bi | Ai | 1 | 0 | 0 | 0 | 1 | 0 | 1 | 0 | 0 |
| +6 | Ai | Ci | Ai | 1 | 0 | 0 | 0 | 0 | 1 | 1 | 0 | 0 |
| −5 | Bi | Ci | Bi | 0 | 1 | 0 | 0 | 0 | 1 | 0 | 1 | 0 |
| +4 | Bi | Ai | Bi | 0 | 1 | 0 | 1 | 0 | 0 | 0 | 1 | 0 |
| −9 | Ci | Ci | Ai | 0 | 0 | 1 | 0 | 0 | 1 | 1 | 0 | 0 |
| +8 | Ci | Ci | Bi | 0 | 0 | 1 | 0 | 0 | 1 | 0 | 1 | 0 |
| −7 | Ai | Ai | Bi | 1 | 0 | 0 | 1 | 0 | 0 | 0 | 1 | 0 |
| +9 | Ai | Ai | Ci | 1 | 0 | 0 | 1 | 0 | 0 | 0 | 0 | 1 |
| −8 | Bi | Bi | Ci | 0 | 1 | 0 | 0 | 1 | 0 | 0 | 0 | 1 |
| +7 | Bi | Bi | Ai | 0 | 1 | 0 | 0 | 1 | 0 | 1 | 0 | 0 |

Using the ISVM control method, the required stator winding voltage and PDL current were generated from the active and zero vectors. Figure 8 shows the active vectors produced from the previously presented combinations.

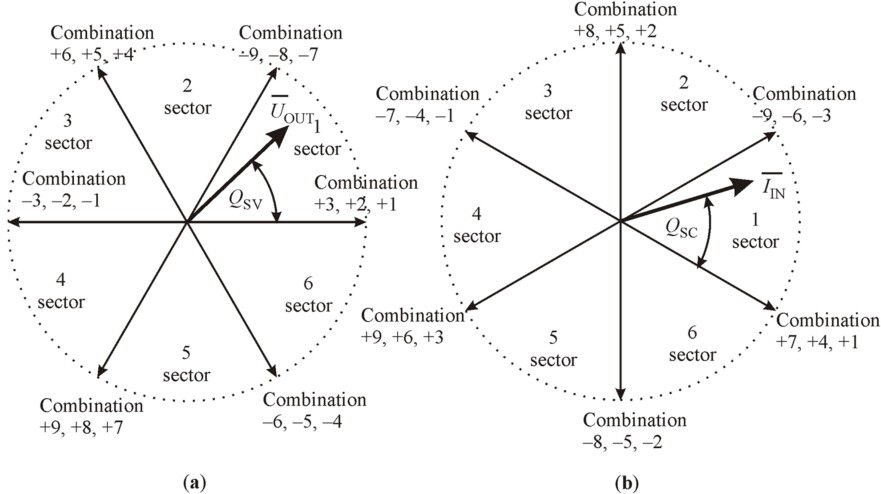

(a)                                                                                       (b)

**Figure 8.** Graphics of the active vectors presented in Table 4: (**a**) graphic of the motor stator windings voltage vectors; (**b**) graphic of the PDL current vectors.

The motor stator winding voltage and PDL current are formed from two certain sector vectors. The frequency of MC motor supply voltage is determined by the angle $Q_{SC}$, and the current displacement at the connection to the PDL point is determined by the angle $Q_{SV}$ [7].

$$Q_{SC} = (\omega_i t - \varphi_{in}) + 30°, \tag{13}$$

$$-30° \leq \omega_i t - \varphi_{in} \leq +30°, \tag{14}$$

where $t$ is the time for synchronization with the PDL frequency, and $\varphi_{in}$ is the current displacement angle.

## 4. Application of the AC–AC Matrix VFD as the Reactive Power Compensation Device

To improve the power factor in current 0.4 kV PDLs, it is proposed to replace some of the conventional AC–DC–AC VFDs with the novel AC–AC matrix VFD, as presented in Figure 9.

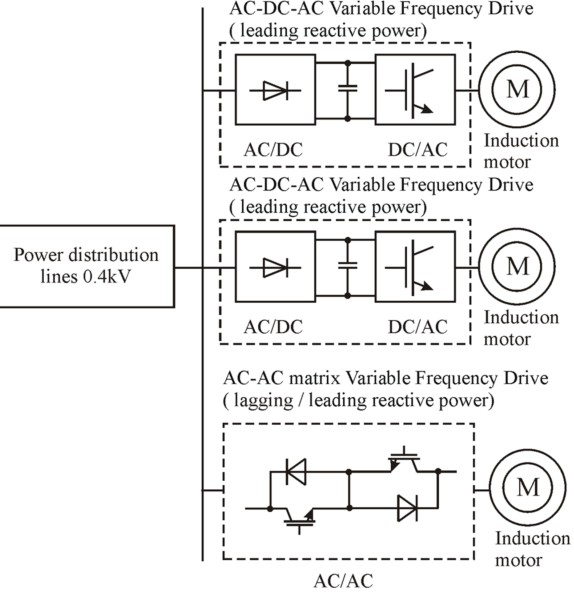

**Figure 9.** The structure of researched system.

As described above, the main difference between conventional AC–DC–AC VFDs and the AC–AC matrix VFD is related to electronic power circuits and control. The conventional AC–DC–AC VFD power circuits consist of a diode supply-side rectifier, an intermediate DC circuit, and an inverter. The AC–AC matrix VFD power circuits consist of nine bidirectional electronic switches based on IGBT. As shown by the theoretical calculations, initial experimental study, and initial simulation in Matlab/Simulink, the conventional AC–DC–AC VFD produces leading reactive power at the point of connection to the PDL. Moreover, theoretical calculations and initial simulations in Matlab/Simulink showed that the AC–AC matrix VFD has the ability to directly control the reactive power produced at the point of connection to the PDL. When these VFDs are connected to the same PDL, the matrix VFD must compensate for the reactive power produced by the conventional AC–DC–AC VFD. It is, therefore, proposed to replace the electronic equipment of one of the conventional VFDs installed in a commercial or industrial building. Instead of three stages (diode supply-side rectifier, intermediate DC circuit, and inverter), nine bidirectional electronic switches are installed, leaving the VFD induction motor the same. In this way, a PDL power factor close to unity can be achieved. By using the AC–AC matrix VFD, the transfer of active power to the motor and the reactive power compensation are performed simultaneously. A Matlab/Simulink simulation study was performed to determine the reactive power compensation range of the system shown in Figure 9.

The input current and motor supply voltage of the MC can be described by the following equations [13]:

$$I_{IN} = \frac{\sqrt{3}}{2} I_{OUT}\, m cos\varphi_{OUT}, \tag{15}$$

$$V_{OUT} = \frac{\sqrt{3}}{2} V_{IN}\, m cos\varphi_{IN}, \tag{16}$$

where $I_{IN}$ is the MC input current, $V_{OUT}$ is the MC motor supply voltage, $m$ is the modulation index, $\varphi_{IN}$ is the input displacement angle, and $\varphi_{OUT}$ is the motor supply displacement angle.

The AC–AC matrix VFD electronic switches were simulated using standard lossless Matlab/Simulink blocks. On the basis of Equations (15) and (16), the power equation of the AC–AC matrix VFD was derived as follows:

$$P_{IN} = \frac{3}{2}V_{PDL}I_{IN}cos\varphi_{IN} = P_{OUT} = \frac{3}{2}V_{OUT}I_{OUT}cos\varphi_{OUT}, \tag{17}$$

where $V_{PDL}$ is the voltage of the PDL, and $f_{PDL}$ is the frequency of the PDL.

From Equation (15), it can be seen that an increase in the induction motor winding current displacement angle $\varphi_{OUT}$ causes a decrease in PDL branch current $I_{IN}$. From Equation (16), it can be seen that an increase in the PDL current displacement angle $\varphi_{IN}$ causes a decrease in induction motor winding voltage $V_{OUT}$.

According to Equation (17), it can be seen that the active power of the AC–AC matrix VFD must be a constant value, thereby ensuring the operation of the powered induction motor. To ensure constant $P_{OUT}$ in the case of variable cos $\varphi_{IN}$, the motor supply voltage $V_{OUT}$ must be regulated within a certain range. To achieve this, constant $P_{OUT}$ can be maintained during the control of the modulation index $m$ (0–1).

For application of the AC–AC matrix VFD as the reactive power compensation device, the lower point of $V_{OUT}$ at induction motor stator windings was determined.

$$V_{OUT} = \frac{V_{IN}}{\sqrt{3}}. \tag{18}$$

The voltage was reduced to extend the modulation limits. To reduce voltage, the stator windings have to be connected in a delta arrangement (Δ).

The voltage ratio of the induction motor to ensure reactive power compensation is as follows:

$$\frac{V_{OUT}}{V_{PDL}} = \frac{1}{\sqrt{3}} = 0.58.$$ (19)

According to Equation (16),

$$\frac{\sqrt{3}}{2} \, mcos\varphi_{IN} = 0.58,$$ (20)

$$mcos\varphi_{IN} = 0.67.$$ (21)

The following equations determine the limits within which m and $\varphi_{IN}$ vary:

$$0.67 < m < 1,$$ (22)

$$0 < \varphi_{IN} < 48^{\circ}.$$ (23)

$$0 < \tan\varphi_{IN} < 1.1.$$ (24)

Equation (21) is valid only if the AC–AC matrix VFD motor supply voltage frequency corresponds to the PDL voltage frequency. In the event that the MC motor supply voltage frequency is not constant, one possible condition is

$$\frac{V_{OUTf}}{f_{OUT}} = const.$$ (25)

In the case of a VFD load with constant torque variation, the following equations apply:

$$V_{outf} = \frac{f_{OUT} \cdot V_{OUT}}{f_{PDL}},$$ (26)

$$\frac{f_{OUT} \cdot V_{OUT}}{f_{PDL}} = \frac{\sqrt{3}}{2} \, V_{IN} \, mcos\varphi_{IN},$$ (27)

$$mcos\varphi_{IN} = 0.67 \times \frac{f_{OUT}}{f_{PDL}}.$$ (28)

Equation (28) shows that the ability to adjust the angle increases if the motor operating frequency $f_{OUT}$ is lower than the frequency of the power distribution line. In the case of an AC–AC matrix VFD load with squared torque variation, the following equations apply:

$$V_{outf} = V_{OUT} \left( \frac{f_{OUT}}{f_{PDL}} \right)^2,$$ (29)

$$mcos\varphi_{IN} = 0.67 \times \left( \frac{f_{OUT}}{f_{PDL}} \right)^2.$$ (30)

As shown in Equation (30), it is possible to further adjust the angle $\varphi_{IN}$ under squared torque load. However, the reactive power is determined not only by the angle $\varphi_{IN}$, but also by the active power $P_{OUT}$ of the AC–AC matrix VFD. As the active power decreases, the reactive power also decreases, unless this is compensated for by an increase in allowable compensation angle $\varphi_{IN}$. Assuming that the AC–AC matrix VFD motor is always loaded at the maximum allowable torque, when the AC–AC matrix VFD operating frequency is varied, the motor current remains at the maximum permissible $I_{OUT}$.

$$P_{OUT} \sim \frac{f_{OUT}}{f_{PDL}} V_{OUT} I_{OUT}.$$ (31)

Keeping the active power at a nominal speed of 100%, the reactive power of the AC–AC matrix VFD at variable speed was calculated. On the basis of Equations (28) and (30), the

compensation angles $\varphi_{IN}$ were calculated. The $\tan\varphi_{IN}$ and the reactive power $Q$ required for compensation were calculated on the basis of the compensation angles.

The calculation results presented in Figure 10 show that, as the frequency of the AC–AC matrix VFD decreases, the reactive power increases. Thus, a decrease in the operating frequency of the VFD increases the possibility of reactive power compensation. Reactive power compensation is available over the entire operating frequency range of the AC–AC matrix VFD.

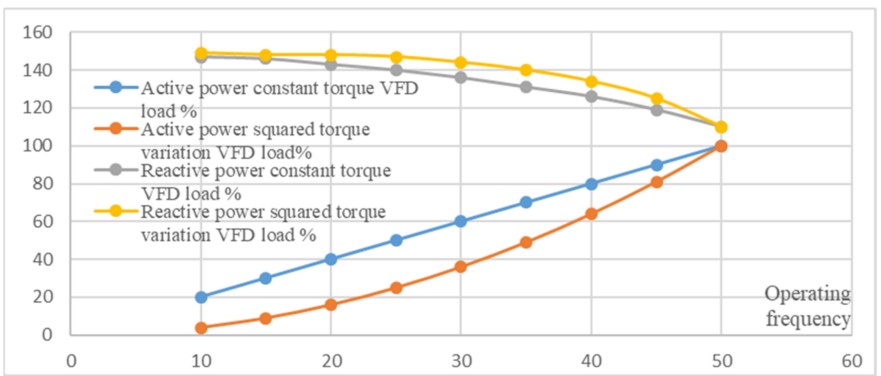

**Figure 10.** Relative active and reactive power at the point of AC–AC matrix VFD connection to the PDL as a function of the VFD operating frequency in the cases of VFD load with constant torque and squared torque variation.

### 4.1. Structure of the Matlab/Simulink Model for Simulating the AC–AC Matrix VFD

The Matlab/Simulink models of the reactive power compensation systems consisted of three pieces (Figure 11).

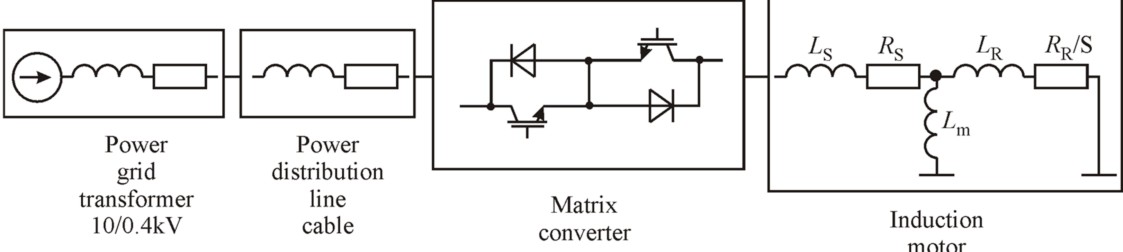

**Figure 11.** Structure of the AC–AC matrix VFD Matlab/Simulink model.

The first part consisted of PDL simulation blocks that simulated 10/0.4 kV transformer and PDL cables using the following standard Matlab/Simulink blocks: ideal sinusoidal AC voltage source and three-phase series RLC branch.

The second part consisted of Matlab/Simulink blocks for simulating MC power circuits (Figure 12) using standard blocks simulating an IGBT device in parallel with a series RC snubber. In the second part, space vector modulation control modules were also attached [14]. Subsystems running Park and Clarke transformations were used for this purpose. The sector separation subsystem was implemented using standard Matlab/Simulink logic and comparison blocks.

The third part consisted of Matlab/Simulink blocks for simulating the MC-powered induction motor according to an equivalent diagram of the induction motor [15] using standard blocks simulating a three-phase series RL branch.

For recording the active and reactive power data, the PLL-driven, positive-sequence block was used to record the positive-sequence active power P (W) and reactive power Q (var) of a periodic set of voltages and currents.

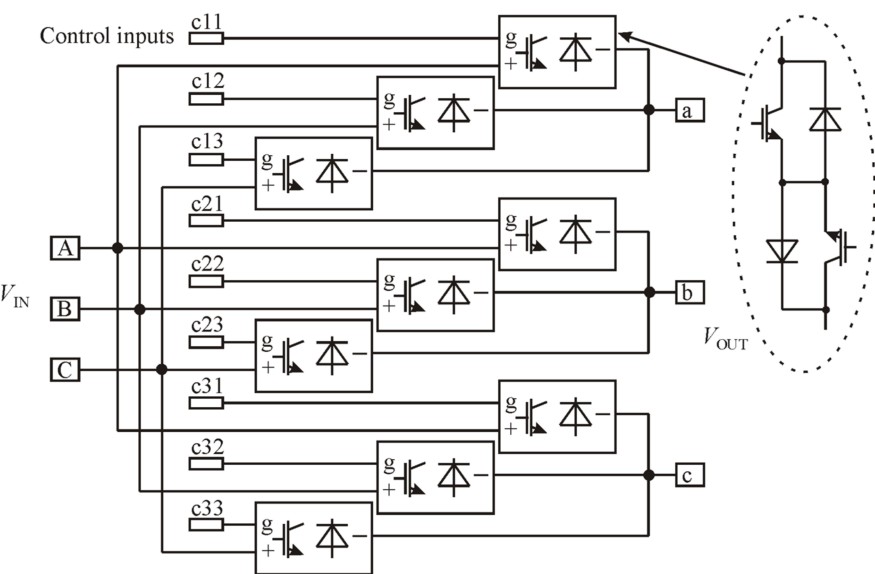

**Figure 12.** Structural diagram of the nine ideal bidirectional switches in Matlab/Simulink model.

### 4.2. Structure of the Matlab/Simulink Model for Simulating the Conventional AC–DC–AC VFD

The Matlab/Simulink model of the conventional AC-DC-AC VFD consisted of four parts. The first part of this model used a standard Matlab/Simulink block to simulate the full-bridge three-phase diode rectifier connected to the PDL using a universal bridge mask Matlab/Simulink block, in which the "power electronic device—diode" settings were applied. The second part implemented the capacitor bank of the DC circuit using a parallel RLC branch Matlab/Simulink block, in which the "branch type—C" settings were applied. The third part consisted of a Matlab/Simulink block implementing an IGBT full-bridge three-phase inverter using a universal bridge mask Matlab/Simulink block, in which the "power electronic device—IGBT/diode" settings were applied. The "PWM generator 2 level" block was also used to control the inverter. The fourth part consisted of Matlab/Simulink blocks for simulating the AC–DC–AC powered induction motor according to an equivalent diagram of the induction motor using standard blocks simulating a three-phase series RL branch.

### 4.3. Reactive Power Compensation Limits of the AC–AC Matrix VFD

The reactive power produced by the MC-based VFD depends directly on the current displacement angle at the point of connection to the PDL. This compensation range was determined by way of Matlab/Simulink simulations. The values of reactive power were determined by keeping the real power transferred to the induction motor constant. As a result, changes in real power caused by current displacement angle changes (rectifier stage) were compensated for by voltage modulation index correction (inverter stage).

These ranges of reactive power compensation are quite wide because an extension of these limits was applied. While the VFD's MC was connected to a 400 V line-to-line PDL, the induction motor windings were connected in a "delta" arrangement and powered by 230 V line-to-line voltage. This enabled changing the ISVM modulation index within a wider range. Consequently, at a displacement angle of zero, the modulation index was significantly reduced. However, when the displacement angle increased, there was a much wider voltage correction range due to the delta connection.

The simulation results presented in Figure 13 clearly show that, using the MC-based VFD, the reactive power can be compensated for in both directions (leading and lagging), depending on the current displacement angle.

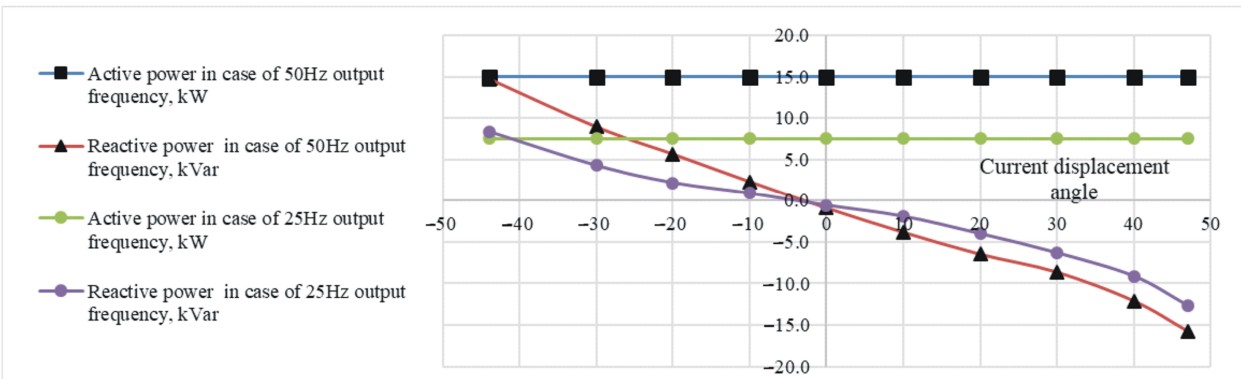

**Figure 13.** The reactive power compensation ranges of the 15 kW MC-based VFD connected to 0.4 kV PDL.

The simulation results presented in Figure 13 show that the possible displacement angles ranged from $-44°$ to $+47°$. MC-based VFD control in this angle range allowed the real power transferred to the motor to be kept constant. For the 15 kW induction motor, the reactive power produced at these displacement angles varied from the leading 14.7 kVar to the lagging 15.7 kVar. Increasing the displacement angle above this specified range led to a decrease in the voltage at the induction motor stator windings and a decrease in the real power transferred to the induction motor.

As mentioned previously, the voltage drop due to the displacement angle change has to be recovered by increasing the value of the modulation index. Upon reaching the limits determined by the displacement angle specified range, the modulation index reaches its maximum value (shown in Table 5) and cannot recover a voltage drop caused by a further increase in the current displacement angle.

**Table 5.** Reactive power at the point of the VFD connection to the PDL.

| Number of Novel AC–AC Matrix VFDs | Total Power of Novel AC–AC Matrix VFDs, kW | Number of Conventional AC–DC–AC VFDs | Total Power of Conventional AC–DC–AC VFDs, kW | Reactive Power Produced by the Number Conventional AC–DC–AC VFDs, kVar | Reactive Power at the Point of VFD Connection to PDL, kVar | Displacement Angle in the AC–AC Matrix VFD, ° | Modulation Index in the AC–AC Matrix VFD |
|---|---|---|---|---|---|---|---|
| 1 | 15 | 1 | 15 | −3.37 | 0 | −13 | 0.72 |
| 1 | 15 | 2 | 30 | −6.21 | 0 | −25 | 0.79 |
| 1 | 15 | 3 | 45 | −8.56 | 0 | −30 | 0.81 |
| 1 | 15 | 4 | 60 | −10.55 | 0 | −34 | 0.85 |
| 1 | 15 | 5 | 75 | −12.11 | 0 | −38 | 0.92 |
| 1 | 15 | 6 | 90 | −13.36 | −0 | −40 | 0.96 |
| 1 | 15 | 7 | 105 | −14.40 | −0.34 | −42 | 1.00 max value |

These simulation results were confirmed by mathematical calculation according to Equation (21). The influence of the indirect space vector modulation control method was mathematically evaluated. The modulation index calculated to compensate for the voltage drop across the delta-connected induction motor windings was calculated using Equation (21) as presented in Figure 14.

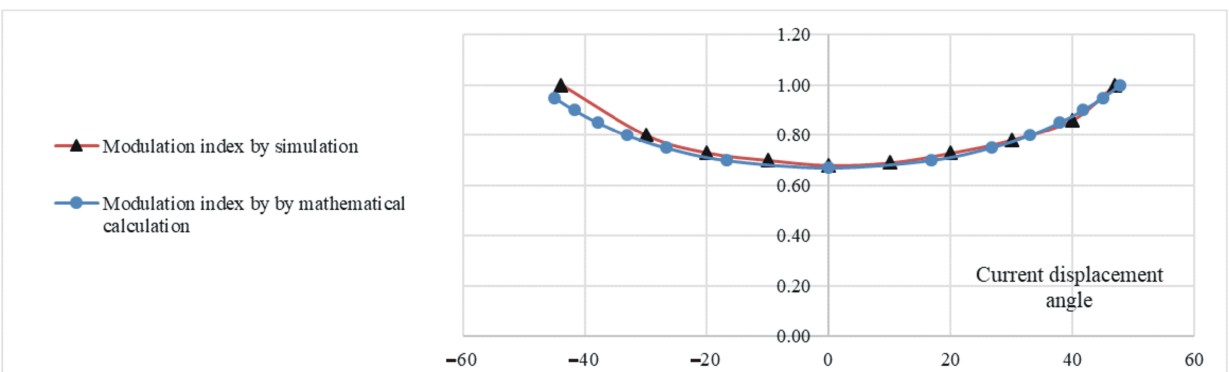

**Figure 14.** ISVM modulation index for voltage drop recovery.

### 4.4. Compensation of Reactive Power Produced by Conventional AC–DC–AC Variable Frequency Drives

In order to evaluate the ability of the AC–AC matrix VFD to compensate the reactive power produced by conventional AC–DC–AC VFDs and keep the power factor close to unity, a Matlab/Simulink model was used. As shown in Figure 15, the model included a 0.4 kV PDL branch to which one AC–AC matrix VFD and a number of conventional AC–DC–AC VFDs were connected. The 15 kW induction motors were powered by the MC converter and by conventional AC–DC–AC frequency converters.

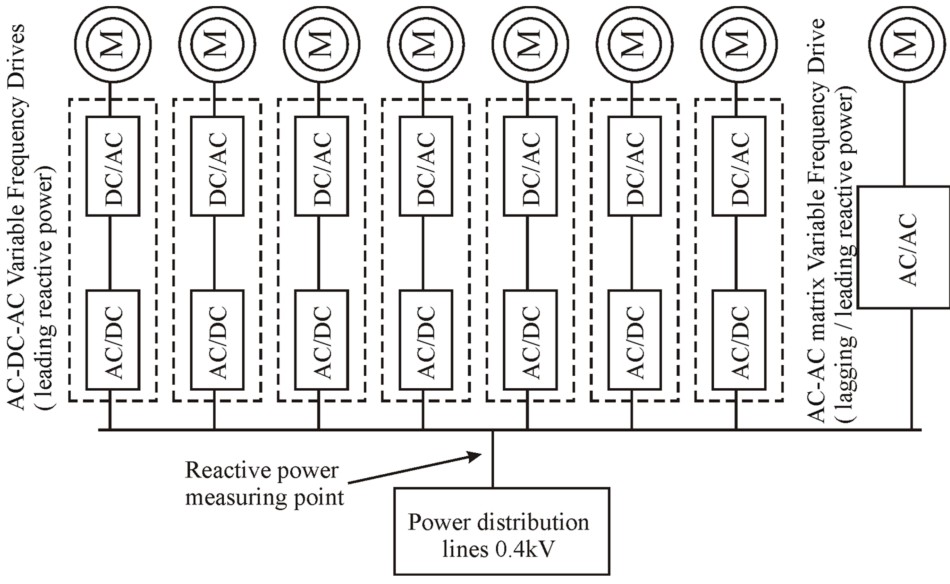

**Figure 15.** The researched system for conventional AC–DC–AC VFD reactive power compensation.

In the course of the study, the number of conventional AC–DC–AC VFDs was varied from one to seven. To compensate the leading reactive power in the PDL produced by the conventional AC–DC–AC VFDs, the current displacement angle of the AC–AC matrix VFD was regulated within the permissible limits.

As mentioned above, an increase in the current displacement angle caused a decrease in the voltage of the MC-powered induction motor stator winding and a decrease in transferred active power. This is directly related to ISVM control, which is applied as the main control strategy for AC–AC matrix VFDs. To recover this decrease, the modulation index was increased. This allowed the active power that was transferred to the induction motor by the MC constant to be maintained. However, the Matlab/Simulink simulation shows that the limits for the MC motor supply voltage recovery by the modulation index were not wide enough. As shown in Table 5, the current displacement angle and modulation index of the ISVM had to be significantly increased as the number of conventional AC–DC–AC

VFDs increased. When connecting more than six conventional AC–DC–AC VFDs, the modulation index had to be increased to the maximum allowed value. The unchanged active power transferred to the induction motor was maintained. However, under such conditions, the AC–AC matrix VFD could no longer fully compensate for the reactive power and maintain the unity power factor. Therefore, it can be argued that an ISVM-controlled AC–AC matrix VFD can only compensate for the reactive power produced by up to six conventional AC–DC–AC VFDs.

The relationship between the reactive power produced by the AC–DC–AC VFD at the point of VFD connection to the PDL and the motor load torque was investigated using the Matlab/Simulink simulation. This relationship was formed at different motor stator winding voltage frequencies and different rotor speeds. The simulation results are shown in Figure 16. Accordingly, it can be stated that VFD produces leading reactive power with varying motor load torque and rotor speed. The value of this reactive power depends directly on the motor load torque and rotor speed. This assumes that the proportions between the number of novel AC–AC matrix VFDs connected to the PDL and the number of conventional AC–DC–AC VFDs will remain similar to those described in Table 5 to maintain the power factor close to unity with varying motor load torque and rotor speed.

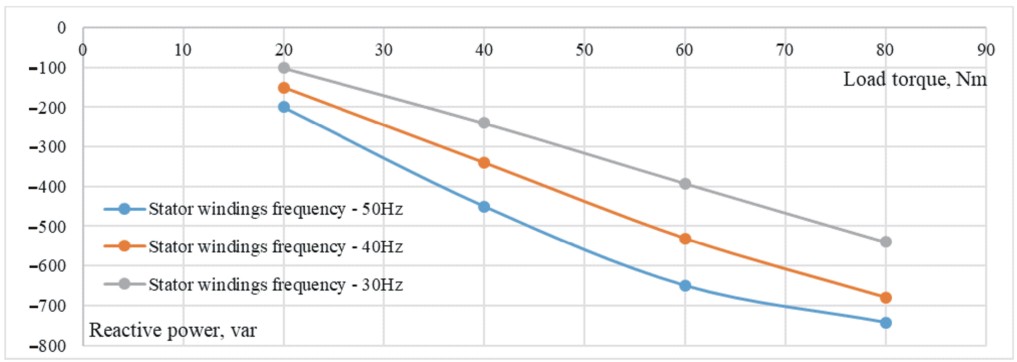

**Figure 16.** The reactive power at the point of single AC–DC–AC VFD connection to the PDL as a function of motor load torque.

## 5. Discussion

The idea to conduct this study arose from the observation of reactive power compensation problems in large building PDLs. It was observed that the interaction of a number of conventional AC–DC–AC VFDs with a conventional capacitor-based local compensation device leads to overcompensation in the 0.4 kV PDLs of the Kaunas Arena building. Capacitive reactive power compensators are installed in this building. This is due to the fact that conventional compensation devices are designed to compensate for the lagging reactive power produced by inductive loads, such as AC motors. This highlighted the demand for the compensation of capacitive reactive power that is not predicted by the designer.

Theoretical calculations show that a conventional AC–DC–AC VFD generates leading reactive power equal to about 30% of real power. Experimental studies confirmed this. This leading reactive power is produced by both single-phase and three-phase supply AC–DC–AC VFDs. Thus, this study is relevant and timely, as confirmed by the example provided above.

Energy companies have proposed installing large quantities of inductive reactors for power factor improvement. First of all, this is not ecologically acceptable due to the high consumption of nonferrous metals. Another shortcoming would be a significant financial cost for the company operating the Kaunas Arena building. The use of AC–AC matrix VFDs would avoid both of these disadvantages.

An AC–AC matrix VFD operating as the reactive power compensation device has another obvious advantage over capacitor-based compensation devices, namely there is no need to use large capacitors. This is an advantage in terms of volume and reliability.

Another important application of the AC–AC matrix VFD is the voltage regulation of the distribution grid. Due to the rapidly evolving solar energy, the distribution grid faces the problem of voltage fluctuations, which can be solved by using the AC–AC matrix VFD as a static reactive power compensator. A particularly important feature of this compensator is that it can generate both lagging and leading power. Future studies will investigate the application possibilities of AC–AC matrix VFD bidirectional reactive power in detail.

## 6. Conclusions

Due to its specific properties, a conventional AC–DC–AC VFD generates leading reactive power. From a technical point of view, this reactive power can be compensated for by induction reactors. However, the high cost, weight, and dimensions of these reactors make this method of compensation unusable. As a result, it was proposed to replace a certain number of conventional AC–DC–AC VFDs with AC–AC matrix VFDs. Studies have shown that, in this way, it is possible to bring the power factor closer to unity.

The application of indirect space vector modulation provides an opportunity to control the current displacement angle of the MC independently from the voltage of the PDL. As a result, the AC–AC matrix VFD can be applied for reactive power compensation. It was determined that the AC–AC matrix VFD can provide the appropriate amount of both leading and lagging reactive power. This feature assumes the application of this AC–AC matrix VFD as the reactive power compensation device.

An increase in the current displacement angle causes a decrease in the voltage of the stator windings and active power transferred to the induction motor. To recover this voltage decrease, the modulation index of the ISVM has to be increased. The determined applicable displacement angle ranges from $-44°$ to $+47°$. To maintain constant power transfer to the motor, the range of the current displacement must not exceed the determined range.

The AC–AC matrix VFD has the ability to compensate for the reactive power produced by conventional AC–DC–AC VFDs that capacitor-based local reactive power compensation devices cannot compensate for. A single AC–AC matrix VFD has the ability to compensate for the reactive power produced by six conventional AC–DC–AC VFDs with the same nominal power.

The compensation power value of the innovative AC–AC matrix VFD is independent from the drive load and frequency. This AC–AC matrix VFD can perform reactive power compensation, even when the VFD operation mode changes.

**Author Contributions:** Conceptualization, G.P.; methodology, G.S.; software, G.P.; validation, G.S. and G.P.; formal analysis, G.S.; investigation, G.S. and G.P.; resources, G.P., G.S. and A.J.; data curation, G.S.; writing—original draft preparation, G.P. and G.S.; writing—review and editing, A.J. and A.G.; visualization, G.P.; project administration, A.J.; funding acquisition, A.J. All authors have read and agreed to the published version of the manuscript.

**Funding:** This research was supported by the European Economic Area (EEA) and Norway Financial Mechanism 2014–2021 under Grant EMP474.

**Conflicts of Interest:** The authors declare no conflict of interest.

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
