# Peer review of "Application of Novel AC–AC Matrix VFD for Power Factor Improvement in Conventional AC–DC–AC VFD-Loaded Power Distribution Lines"

_electronics, doi:10.3390/electronics11070997_

Round 1

Reviewer 1 Report

This manuscript focuses on AC-AC matrix VFD for improving power factor in an AC-DC-AC VFD loaded distribution line. The theoretical concept is fairly good. The mathematics and illustrations are reasonable. The overall organization of the paper is fine.

Authors are suggested to revise the manuscript thoroughly to correct a few syntactical mistakes. Some cross-referencing have issues in the document and it shows as errors in the PDF. Also, the technical writing can be improved. Authors should be careful about these aspects of a good journal article.

More simulation results are expected. No step change has been shown in tests. Loading effects have not been illustrated in transient simulations for power factor (PF) amelioration or loss minimization. In addition, some comparative premises can be included to underscore the superiority of the presented scheme over the existing VFD-loaded PF improvement methods.

Thank you.

Reviewer 2 Report

A new approach to matrix-converter-based AC-AC variable frequency drives (VFD) is presented in this paper. The general idea of the paper seems to be good. However, I have several concerns that should be properly addressed by the authors, and they are as follows:

1) Mention the major contributions of this paper at the end of the Introduction section.

2) Importance of variable frequency drives (VFD) technology can be added in the Introduction section.

3) How the angle alpha has been determined in this work.

4) It is better to support some equations with some proper references or convincing proof.

5) Expand the results section with proper discussion.

6) Proposed structure in figure 9 can be explained in a better way.

7) The conclusion has to be improved to show the power of the proposed model.

8) In page 11, some references are missing. Please correct them.

Reviewer 3 Report

Dear Authors,

This is very interesting manuscript and have detailed demonstration of the novel ideas. My specific feedback or recommendation is to that your manuscript require extensive English proof-reading editing. 

Author Response

Dear reviewer.

My manuscript passed the extensive English proof-reading editing.

Round 2

Reviewer 1 Report

The revised manuscript addresses the recommendations quite fine.

Thank you.